# An allocentric human odometer for perceiving distances on the ground plane

**Liu Zhou[1], Wei Wei[1,2], Teng Leng Ooi[2]\*, Zijiang J He[1]\***

[1]Department of Psychological and Brain Sciences, University of Louisville, Louisville, United States; [2]College of Optometry, The Ohio State University, Columbus, United States

**Abstract** We reliably judge locations of static objects when we walk despite the retinal images of these objects moving with every step we take. Here, we showed our brains solve this optical illusion by adopting an allocentric spatial reference frame. We measured perceived target location after the observer walked a short distance from the home base. Supporting the allocentric coding scheme, we found the intrinsic bias , which acts as a spatial reference frame for perceiving location of a dimly lit target in the dark, remained grounded at the home base rather than traveled along with the observer. The path-integration mechanism responsible for this can utilize both active and passive (vestibular) translational motion signals, but only along the horizontal direction. This asymmetric path-integration finding in human visual space perception is reminiscent of the asymmetric spatial memory finding in desert ants, pointing to nature's wondrous and logically simple design for terrestrial creatures.

## eLife assessment

This **important** study reveals the use of an allocentric spatial reference frame in the updating perception of the location of a dimly lit target during locomotion. The evidence supporting this claim is **compelling**, based on a series of cleverly and carefully designed behavioral experiments. The results will be of interest not only to scientists who study perception, action and cognition but also to engineers who work on developing visually guided robots and self-driving vehicles.

## Introduction

When viewing natural scenes, we readily appreciate the visual space expanse that 'fills' the scene being enveloped by the sky above and terrain below our feet (*Figure 1a and b*). Notably, within this phenomenological visual space, our visual system employs the prevalent ground surface, where creatures and objects frequently interact, as a reference frame for coding spatial locations (*Gibson, 1950*; *Sedgwick, 1986*; *Sinai et al., 1998*). Empirical findings have revealed the ground-based spatial coding scheme accurately localizes objects when the horizontal ground surface is continuous and carries rich depth cues (*Thomson, 1983*; *Rieser et al., 1990*; *Loomis et al., 1992*; *Loomis et al., 1996*; *Sinai et al., 1998*; *Meng and Sedgwick, 2001*; *Wu et al., 2004*; *He et al., 2004*; *Bian et al., 2005*). Attesting to the significant role of the ground surface, it has been found that object localization becomes inaccurate when the ground surface is disrupted by a gap or an occluding box (*Sinai et al., 1998*; *He et al., 2004*). Similarly, object localization is inaccurate in the dark when the ground is not visible (*Ooi et al., 2001*; *Ooi et al., 2006*; *Philbeck and Loomis, 1997*). In fact, research conducted in total darkness to prevent the visual system from obtaining depth information from the ground reveals the visual system defaults to using an implicit semi-elliptical curved surface representation that we refer to as the *intrinsic bias* to localize objects (*Ooi*

\*For correspondence:
ooi.22@osu.edu (TLO);
zjhe@louisville.edu (ZJH)

Competing interest: The authors declare that no competing interests exist.

*et al., 2001*; *Ooi et al., 2006*; *Gogel and Tietz, 1979*). *Figure 1c* illustrates the intrinsic bias' semi-elliptical representation in the dark (dashed white curve) that leads to an observer perceiving a dimly lit target (green ring) at the intersection (green disc) between its projection line from the eye and the intrinsic bias. Furthermore, when a parallel row of texture elements on the ground serves as the depth cue (*Figure 1d*), object localization becomes more accurate. It is as if the objects are now located along a curved reference (solid white curve), which is less slanted than the intrinsic bias. This suggests the intrinsic bias, acting as the prototype spatial reference of the ground surface when visual cues are not visible, integrates with the visible depth cues such as texture gradients on the ground to form a new ground surface representation (*Wu et al., 2015*). In this way, the ground surface representation depends on the weighted contributions of the depth information on the ground and the intrinsic bias.

Nevertheless, current understanding of the ground-based spatial coding scheme is limited to situations where the tested observer stood still. Namely, just as in *Figure 1c*, now redrawn in *Figure 2* and referred to as the *baseline-stationary* condition, the observer is instructed to stand still at one location during the test, which anchors his/her intrinsic bias on the ground under his/her feet. In this way, object location in visual space can be computed relative to the intrinsic bias. However, because retinal images of the stationary environment move when one moves (e.g., walk), it is unclear what additional computation the ground-based spatial coding scheme requires to construct a stable perceptual space. We proposed that during translational movements, such as walking, the brain anchors the intrinsic bias at one location on the ground surface using a *path-integration* mechanism (*Mittelstaedt and Mittelstaedt, 1980*; *Wehner and Srinivasan, 1981*; *Rieser, 1989*; *Etienne et al., 1996*; *McNaughton et al., 1996*; *Loomis et al., 1999*; *Mittelstaedt and Mittelstaedt, 2001*; *Loomis et al., 2013*). The path-integration mechanism generates an estimate of the observer's current position relative to their original location (home base) by integrating each step of traveled length and direction (vector). For example, when the observer walks forward in the dark, the path-integration mechanism estimates the walked length by integrating each step length based on inputs from the vestibular and/or proprioception systems. The intrinsic bias accordingly shifts behind the body over a distance equaling the estimated walked distance; effectively placing the intrinsic bias at the home base should the estimated walked distance be accurate. This leads the visual system to construct the ground surface representation from the (allocentric) intrinsic bias that is fixed to the ground surface at the home base. By doing so, the visual system can obtain a stable allocentric ground surface representation. Accordingly, the ground-based reference frame can use an allocentric, that is, world-centered, coordinate system to code locations in visual space.

The allocentric hypothesis predicts that during self-motion, the intrinsic bias is fixed to the ground location before the motion begins (home base) and remains at the same ground location during self-motion. This hypothesis stands in contrast to an alternative egocentric coordinate system hypothesis, which predicts the intrinsic bias moves along with the observer's body. To distinguish between these two hypotheses, we began by measuring the intrinsic bias when the observer walked in the dark. Consider the scenario where an observer walks from his/her home base (blue cross added in figure for illustrative purpose) in the dark and stops at a new location (*Figure 2b*; red cross added for illustration). According to the allocentric hypothesis, the intrinsic bias (blue) would remain at the home base. Thus, when he/she stops and is presented with a dimly lit test target (illustrated by the green ring), the observer would perceive the target to be nearer and higher (*Figure 2b*) than if he/she had not walked from the home base (*Figure 2a*; we shall dub this the *baseline-stationary* condition). In contrast, if the intrinsic bias had moved along with the observer to the new location (illustrated as the gray curve in *Figure 2c*), as predicted by the alternative egocentric hypothesis, the observer would perceive the target at the same location as that in the *baseline-stationary* condition (*Figure 2a*).

We have premised the allocentric hypothesis on the ground-based spatial coding scheme employing the path-integration (or spatial updating) system to maintain the intrinsic bias at the home base (blue cross in *Figure 2b*). Previous studies have found that when performing spatial memory tasks, in conditions where visual cues are not visible in the dark, the path-integration mechanism computes the traveled distance based on (non-visual) idiothetic distance information, such as from the vestibular, proprioceptive, and efference copy of motor control signals (*Mittelstaedt and Mittelstaedt, 1980*; *Etienne et al., 1996*; *Loomis et al., 1999*; *Loomis et al., 2013*). Thus, we extended this study to characterize the behaviors of the path-integration mechanism in a visual spatial perception

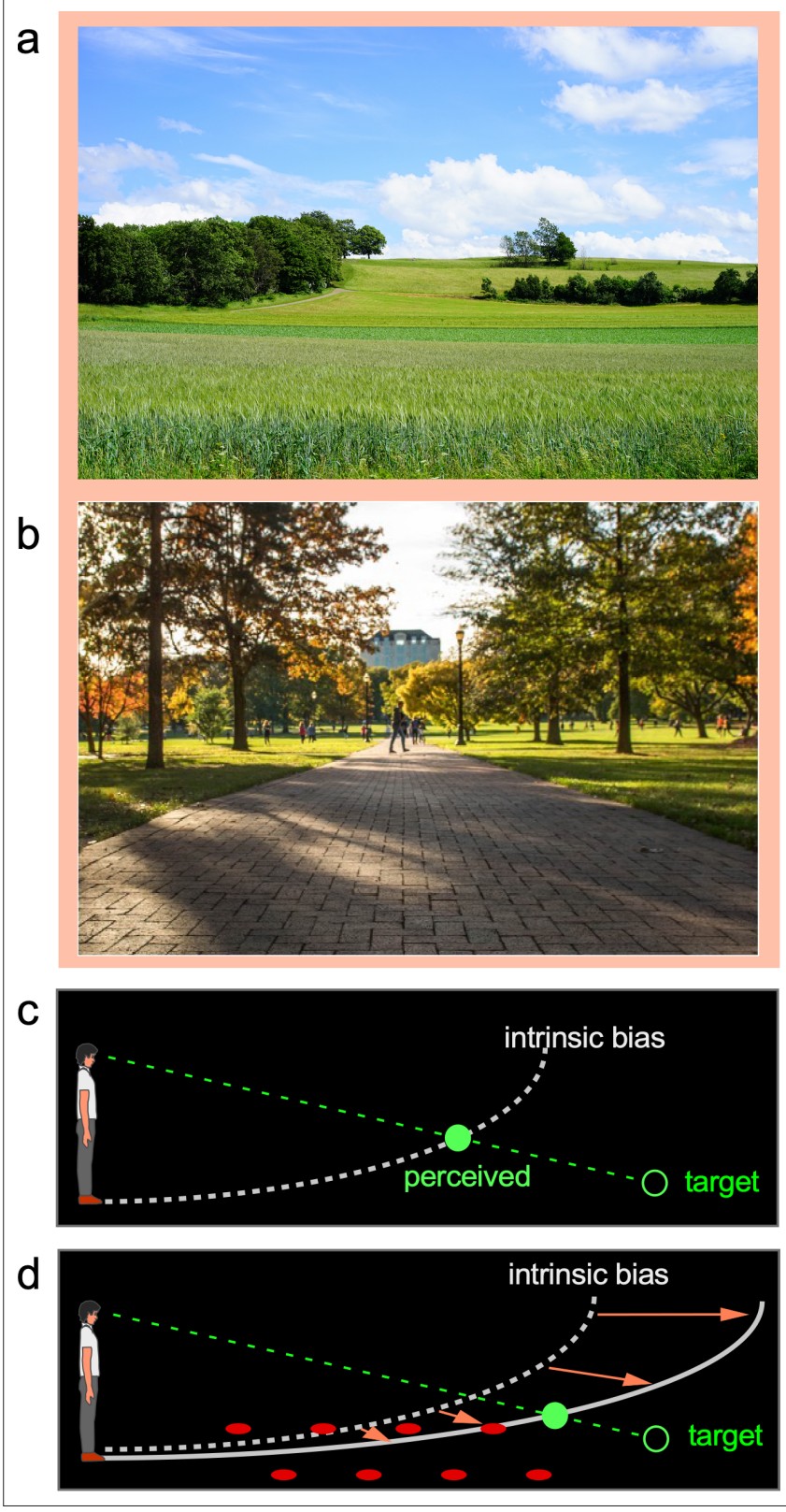

**Figure 1.** The ground-based spatial coding scheme. (**a**) and (**b**) Natural scenes are largely enveloped by the prevalent sky above us and the ground on which we stand. Evidence suggests target locations are judged with respect to the ground surface, which is our terrestrial niche. (**c**) The visual system uses the intrinsic bias, an implicit semi-elliptical surface representation in the dark (illustrated as dashed white curve), to locate a dimly lit target

*Figure 1 continued on next page*

*Figure 1 continued*

(illustrated as unfilled green disc). The target is perceived (filled green disc) at the intersection between the projection line from the eye to the target and the intrinsic bias. Essentially, the intrinsic bias acts like the visual system's internal model of the ground surface. (**d**) When the ground becomes visible, the intrinsic bias integrates with the visible depth cues to form a ground surface representation, which serves as a reference frame to code target location. For example, the parallel rows of texture elements on the ground (illustrated as filled red circles) provide the depth cue for the visual system to construct a ground surface representation (solid white curve) that is less slanted than the intrinsic bias. This leads to a more accurate target localization than in the dark (**c**).

task, to investigate if the path-integration mechanism behaved in a similar manner as in the spatial memory task.

## Results

### Experiment 1: testing allocentric vs. egocentric hypothesis

We first verified the prediction of the allocentric hypothesis (*Figure 2*) by testing a *baseline-stationary* condition (*Figure 3a*) and a *walking* condition (*Figure 3b*). In both conditions, the observer readied for the experiment by sitting on a chair in the waiting area, which was illuminated by a dimly lit LED light from the ceiling. To ready for a test trial, he/she waited for an audio tone. Upon hearing the tone, he/she stood up and aligned his/her feet at the start point (home base) that faced a black curtain in the direction of the testing area. About 30 s later, the experimenter turned off the LED light in the waiting area and the observer drew the curtain open in the dark to begin the trial.

In the *baseline-stationary* condition, after further waiting at the home base (blue cross) for either 12 or 60 s, the observer saw a briefly presented (1 s) dimly lit target. He/she was instructed to judge the target location and to respond by walking blindly to the remembered target location and gesturing its height after reaching the walked destination (*Thomson, 1983*; *Loomis et al., 1996*; *Ooi et al., 2001*; *Ooi et al., 2006*). (While walking, the observer's right hand glided along a horizontal guidance rope, drawn as the yellow line in *Figure 3a*.) Effectively, this blind walking-gesturing task reveals the perceived target location (*x*: walked distance; *y*: gestured height).

In the *walking* condition (*Figure 3b*), the observer walked blindly from the home base (blue cross) to a new location (red cross). While walking, the observer's right hand glided along a guidance rope and stopped at the new location when he/she made contact with a soft plastic wrap on the rope (1.5 m from the home base). After waiting at the new location for either 12 or 60 s, he/she saw a briefly presented (1 s) dimly lit target. He/she then performed the blind walking-gesturing task to indicate the perceived target location.

*Figure 3c* shows the average results of the two conditions. With the 12 s waiting period, judged locations in the *walking* condition (filled green triangles) were significantly nearer than the *baseline-stationary* (filled red circles) condition (p < 0.0001; please refer to the Statistical Analysis of Data at the end of Results section for details of all statistical analyses). The two sets of data points are fitted by the same shaped intrinsic bias curve with about 1.35 m horizontal separation, which is close to the initially walked distance from the home base (1.5 m; in the *walking* condition). This result thus confirms the prediction of the allocentric hypothesis (*Figure 2b*). With the 60-s waiting period, judged locations (open triangles) in the *walking* condition had a smaller, though statistically significant, separation from the *baseline-stationary* condition (open circles) (p < 0.01). This suggests if the waiting period was sufficiently long, suggesting the visual memory of the home base decays over time (*Thomson, 1983*; *Loomis et al., 2013*), the visual system automatically resets the intrinsic bias to the observer's (new/current) location, that is, making it the updated home base. *Figure 3e* plots the average judged angular declination as a function of the physical angular declination for all conditions. The slopes of the regression lines obtained with the least squared methods are close unity, suggesting perceived angular declinations of the targets were largely accurate.

The results above demonstrate the action of the path-integration mechanism. It tracks the observer's location with respect to the external world during locomotion and plays a critical role in maintaining an allocentric reference frame. In the *walking* condition, the path-integration mechanism computes the traveled distance relative to the home base. Experiments 2–4 explored the characteristics of the path-integration mechanism.

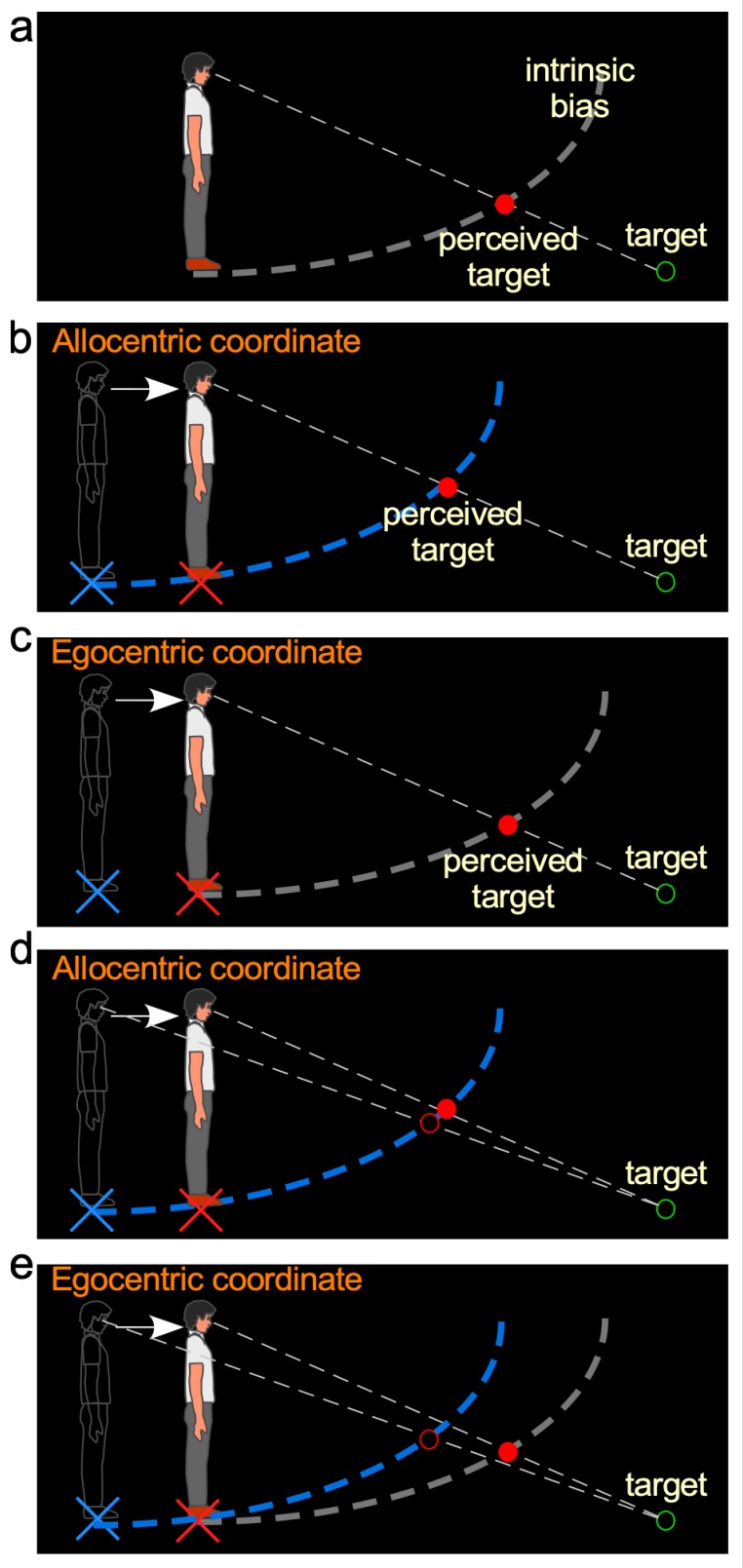

**Figure 2.** Hypotheses and predictions of Experiment 1. (**a**) *Baseline-stationary* condition. A non-moving, static observer perceives the dimly lit target (illustrated as unfilled green disc) at the intersection (illustrated as filled red disc) between the projection line from the eye to the target and the intrinsic bias (dashed white curve). (**b**) The allocentric hypothesis predicts when the observer walks forward from the home base (illustrated as blue cross)

*Figure 2 continued on next page*

*Figure 2 continued*

to a new location (illustrated as red cross), the visual system relies on the path-integration mechanism to keep the intrinsic bias (illustrated as dashed blue curve) at the home base while monitoring the homing error vector (illustrated as white arrow). From the new location, he/she perceives the target (illustrated as unfilled green circle) at the intersection (illustrated as red disk) between the intrinsic bias anchored at the home base and the projection line from the eye to the target. (**c**) The egocentric hypothesis predicts the intrinsic bias tags along with the observer's body (illustrated as gray intrinsic bias) when he/she walks to the new location (illustrated as red cross). From the new location, the target (illustrated as unfilled green circle) is perceived (illustrated as red disk) at the same location as in the *baseline-stationary* condition (in **a**). (**c**) Comparison of judged location of a target (unfilled green circle) before and after walking based (filled vs. unfilled red circles) on the allocentric hypothesis. (**c and d**) Comparison of predicted locations of a target (unfilled green circle) before and after walking (filled vs. unfilled red circles) based on the allocentric and egocentric hypotheses. Note the smaller separation between the filled vs. unfilled red circles in (**c**), suggesting better perceived positional stability with allocentric spatial coding.

## Experiment 2: path integration affected by cognitive load

Since in the *walking* condition (*Figure 3b*), observers were simply attentive to the task at hand but otherwise neutral and not subjected to excessive cognitive demands, one might assume that the path-integration mechanism operates automatically. To test this assumption, we investigated whether the path-integration mechanism requires some attentional resources to function normally (*Amorim et al., 1997*; *Cavanagh et al., 2010*). We thus tested a new, *divided attention walking* condition, where the observer continuously performed a cognitive task, counting number backward, while walking from the home base to the new location. We predicted this would reduce the attention resources available for path integration. For comparison, we also tested the *baseline-stationary* condition as in Experiment 1 (*Figure 3a*).

The average results in *Figure 3d* reveal a small, though significant, difference between the *divided attention walking* condition and the *baseline-stationary* conditions (p = 0.022). Noticeably, the distance underestimation in the *divided attention walking* condition (*Figure 3d*) was much smaller than that found in Experiment 1 (*Figure 3c*). This suggests path integration is less effective when observers were engaged in another mental task that distracted their spatial attention. *Figure 3f* plots the average judged angular declination as a function of physical angular declination for both conditions. The regression lines from the two conditions are similar.

## Experiment 3: path integration from vestibular input

We addressed two issues regarding the idiothetic distance cues used by the path-integration mechanism. First, does it operate only when the observer's movement is self-initiated? Second, does it operate when observer moves in the backward direction? To answer these questions, we moved the observers passively to stimulate their vestibular system (*Mittelstaedt and Mittelstaedt, 2001*; *Israël and Berthoz, 1989*; *Siegle et al., 2009*). That is, instead of instructing the observer to walk in the dark from the home base, he/she stood upright on a rolling-platform that was moved by the experimenter (insets in *Figure 4*). Doing so negated the contributions of the proprioception and motor efference copy information to path-integration while keeping the vestibular information intact.

We first tested a new *vestibular-forward* condition in which the experimenter pushed the observer on the platform forward by 1.5 m to the new location. Once at the new location, the observer stepped down from the platform and stood on the floor to wait for the dimly lit test target to be presented. He/she responded with the blind walking-gesturing task. For comparison, we also tested the *walking* (*Figure 3b*) and *baseline-stationary* (*Figure 3a*) conditions as in Experiment 1. *Figure 4a* reveals the judged target locations in the *vestibular-forward condition* (blue squares) were similar to the *walking* condition (green triangles; p = 0.718), and were significantly nearer than the *baseline-stationary* condition (red circles; p < 0.0001). This indicates the path-integration mechanism can function during passive movements that stimulate the vestibular system.

To answer the second question, we then tested a complementary condition wherein the experimenter pulled the rolling-platform supporting the observer backward by 1.5 m from the home base (*vestibular-backward* condition). As expected, the backward displacement caused judged target locations to be farther than the *baseline-stationary* condition (p < 0.0001; *Figure 4b*). Taken together, the

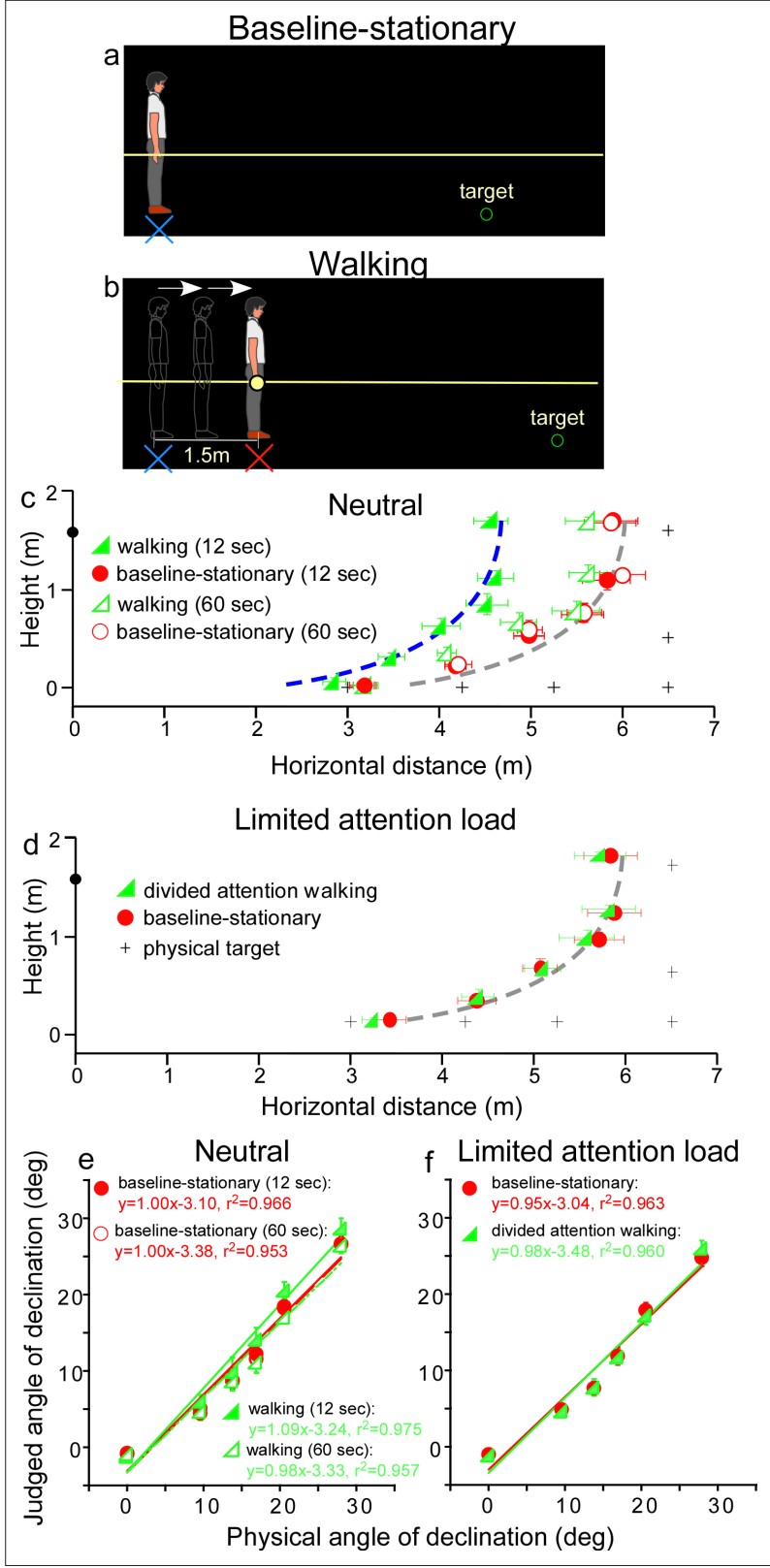

**Figure 3.** Experiment 1. (**a**) *Baseline-stationary* condition: The observer stood at the home base (illustrated as blue cross) and judged the location of a briefly presented dimly lit target. (**b**) *Walking* condition: The observer walked blindly from the home base (illustrated as blue cross) for 1.5 m to the new location (illustrated as red cross). After a short delay (12 or 60 s), the dimly lit target was presented for him/her to judge the location. (**c**) Graph plotting

*Figure 3 continued on next page*

*Figure 3 continued*

the average (*n* = 8) judged target locations from Experiment 1. The black plus symbols, here and in other graphs, represent the physical target locations. The filled and open green triangle symbols represent the results for the 12 and 60 s waiting periods, respectively, in the *walking* condition. The red circle symbols represent the *baseline-stationary* condition. With the 12-s waiting period, judged locations (filled green triangles) in the *walking* condition were significantly nearer than in the *baseline-stationary* (filled red circles) condition. With the 60-s waiting period, judged locations (open triangles) had a much smaller separation from the *baseline-stationary* condition. The blue and gray curves with the same shape are the intrinsic bias fitted to the data by eye. Their horizontal shift is about 1.35 m. The black circle on the vertical axis represents the average eye height of the observers. (**d**) Graph plotting the average (*n* = 8) judged target locations from Experiment 2. The green triangle and red circle symbols represent results from the *divided-attention-walking* and *baseline-stationary* conditions, respectively. Judged locations from both conditions were similar and are fitted by the same intrinsic bias curve. Error bars represent the standard errors of the mean. (**e** and **f**) Average judged angular declination as a function of the physical angular declination. The error bars represent the standard errors of the means among observers (n = 8).

The online version of this article includes the following source data for figure 3:

**Source data 1.** Judged target locations of Experiment 1.

experiment reveals the path-integration mechanism can sufficiently utilize the vestibular cue in both forward and backward moving directions.

*Figure 4c and d* plot the average judged angular declination as a function of physical angular declination. The judged angular declinations are similar for targets with smaller angular declinations that were located farther from the observers. However, for nearer targets, the judged angular declination for the forward and backward conditions had a small difference with respect to the baseline-stationary condition. A similar pattern is also found in *Figure 4e*. Further research, beyond the scope of this paper, is required to investigate this observation.

## Experiment 4: the allocentric principle is not task specific

Previous studies of space perception with stationary observers not undergoing self-motion measured with different response tasks have shown a concordance in finding. Specifically, the perceptual effects found were similar between an action-based task (blind walking-gesturing) and a perception-based task (e.g., verbal reports and perceptual matching) (*Sinai et al., 1998*; *Ooi et al., 2001*; *Wu et al., 2004*; *Zhou et al., 2013*). To confirm that this is also true for observers undergoing self-motion, we repeated the *vestibular-forward* and the *baseline-stationary* conditions in Experiment 3 and employed the verbal report task, wherein the observer verbally reported the perceived eye-to-target distances in feet or meters. We found that verbally reported distances were significantly shorter in the *vestibular-forward* than *the baseline-stationary* condition (p < 0.0001, *Figure 4e*). Thus, both the verbal report (*Figure 4e*) and blind walking-gesturing (*Figure 4a*) tasks show that forward moving in the dark before seeing the dimly lit target can affect judged target distance.

## Experiment 5: horizontal/vertical asymmetry in path integration – a bias for the ground surface

An assumption underlying the allocentric hypothesis is that adoption of the ground-based spatial coding scheme is fitting for our terrestrial existence, where everyday activities, including navigation, are performed with respect to the horizontal ground surface (*Gibson, 1950*). This dependence on the ground surface, that is, an ecological constraint, predicts when the observer travels on a sloping surface in the dark, the visual system would not be able to simultaneously maintain the intrinsic bias at the home base in the vertical and horizontal directions. To investigate this prediction of asymmetric path integration, we tested a new, *stepladder* condition, where the observer descended a stepladder in the dark (*Figure 5a*). As the self-motion here consisted of both horizontal forward and vertical downward vectors, we predicted the intrinsic bias would be spatially updated in the horizontal but not in the vertical direction. This prediction is illustrated in *Figure 5a* where the horizontal coordinate of the intrinsic bias (blue curve) remains at the home base location on the floor while the vertical coordinate travels along with the observer. Thus, upon stepping down from the stepladder and standing on the floor, the observer underestimates the horizontal distance of a dimly lit target. We expected that the target underestimation would be similar to that of a *horizontal-walking* condition (*Figure 5b*).

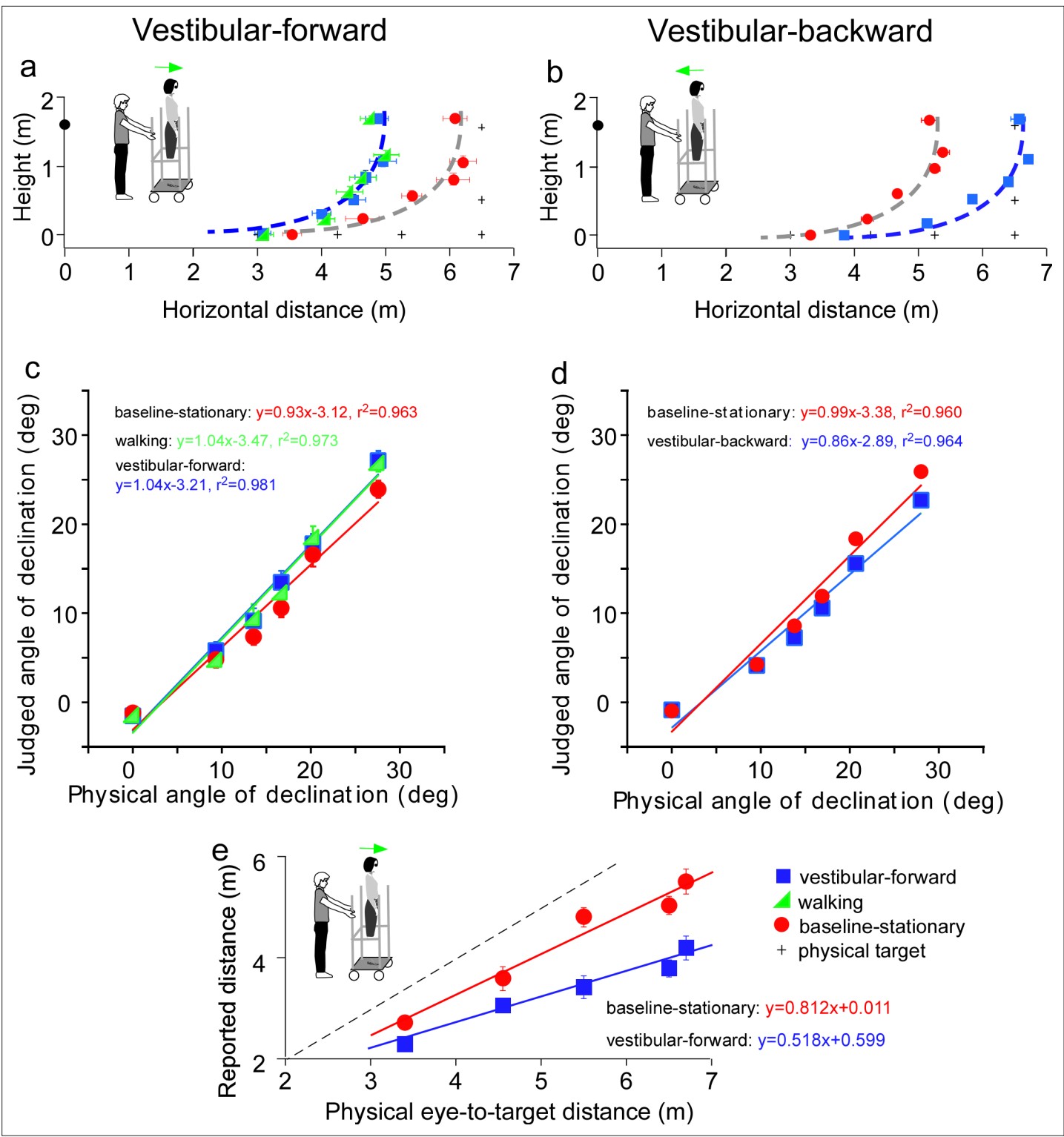

**Figure 4.** Results of Experiment 3 (**a, b**) and Experiment 4 (**e**). (**a**) *Vestibular-forward* condition: The blue square and green triangle symbols represent the *vestibular-forward* and *walking* conditions, respectively, and they show that judged target locations were similar (*n* = 8). These judged target locations were nearer than those from the *baseline-stationary* condition (red circles). (**b**) *Vestibular-backward* condition: Backward translation caused judged target locations (blue squares) to be farther than those from the *baseline-stationary* condition (red circles) (*n* = 8). (**c and d**) Average judged angular declination as a function of the physical angular declination. (**e**) Experiment 4. Verbally reported results of *vestibular-forward* condition show shorter reported eye-to-target distances (blue squares) than in the baseline-stationary (red disc) condition (*n* = 8). Error bars represent standard errors of the mean (n=8).

*Figure 4 continued on next page*

*Figure 4 continued*

The online version of this article includes the following source data for figure 4:

**Source data 1.** Judged target loactions of Experiment 3 and distances of Experiment 4.

In contrast, if the intrinsic bias is spatially updated in both the horizontal and vertical dimensions, the intrinsic bias would remain at the top of the staircase (orange curve, *Figure 5a*). The perceived target location would then be dramatically different from the predicted horizontal-only-updating strategy (*Figure 5b*).

*Figure 5c* depicts the average judged target locations (*n* = 9) from the *stepladder* (blue triangles), *horizontal-walking* (green square) and *baseline-stationary* (red circles) conditions (the *baseline-stationary* condition's setup was the same as in *Figure 3a*). A comparison between the *horizontal-walking* and the *baseline-stationary* conditions reveals the judged horizontal distances were significantly shorter in the *horizontal-walking* condition (p < 0.001). The two sets of data are fitted by the same intrinsic bias profile with a horizontal separation of 1.0 m, which is close to the walked distance (1.06 m) in the *horizontal-walking* condition.

The judged horizontal distances in the *stepladder* condition were significantly shorter than that in the *baseline-stationary* condition (p < 0.001). This confirms the prediction that while descending the stepladder, the visual system spatially updated the horizontal but not the vertical vector of the intrinsic bias (blue intrinsic bias, *Figure 5a*). Further supporting this, we found the data from the *stepladder* and the *horizontal-walking* conditions overlap substantially. *Figure 5d* shows the average judged angular declination as a function of physical angular declination for all three conditions.

Taken together, our experiment revealed when stepping down from a stepladder, the horizontal coordinate of the intrinsic bias is kept at the home base on the floor while the vertical coordinate moves downward with the body. This suggests human's path-integration mechanism, responsible for allocentric spatial coding, operates much more efficiently along the horizontal ground plane than in the vertical direction.

## Statistical analysis of data

### Experiment 1 (*Figure 3c*)

Baseline-stationary condition vs. walking condition with a 12-s waiting period

We applied two-way analysis of variance (ANOVA) with repeated measures (2 test conditions × 6 horizontal distances) to the walked horizontal distance data. The analysis reveals:

- Main effect of test condition (*baseline-stationary vs. walking*): $F_{(31, 7)} = 439.133$, p < 0.001
- Main effect of horizontal distance: $F_{(5, 35)} = 158.676$, p < 0.001
- Interaction effect: $F_{(5, 35)} = 13.255$, p < 0.001

Baseline-stationary condition vs. walking condition with a 60-s waiting period

We applied two-way ANOVA with repeated measures (2 test conditions × 6 horizontal distances) to the walked horizontal distance data. The analysis reveals:

- Main effect of test condition (*baseline-stationary vs. walking*): $F_{(1, 7)} = 13.406$, p < 0.01
- Main effect of horizontal distance: $F_{(5, 35)} = 133.358$, p < 0.001
- Interaction effect: $F_{(5, 35)} = 1.555$, p = 0.198

### Experiment 2 (*Figure 3d*)

Baseline-stationary condition vs. divided attention walking condition

We applied two-way ANOVA with repeated measures (2 test conditions × 6 horizontal distances) to the walked horizontal distance data. The analysis reveals:

- Main effect of test condition (*baseline-stationary vs. divided attention walking*): $F_{(1, 7)} = 8.529$, p = 0.022
- Main effect of horizontal distance: $F_{(5, 35)} = 63.780$, p < 0.001

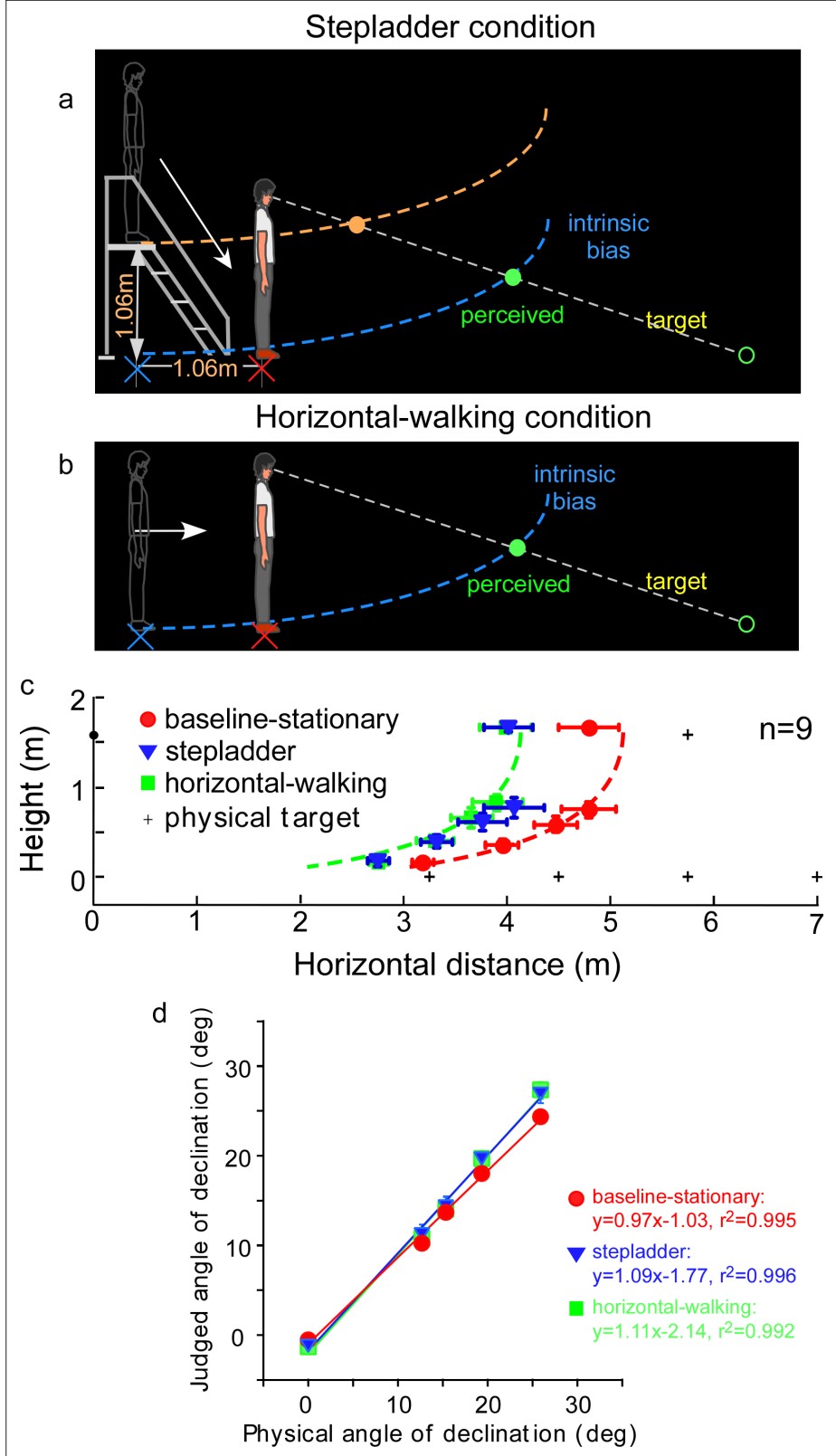

**Figure 5.** Experiment 5: predictions and average results. (**a**) *Stepladder* condition where the observer descends from the stepladder in the dark. If the path-integration process only gauges the horizontally traveled distance, the intrinsic bias would be path integrated in the horizontal but not vertical direction. As such, the horizontal coordinate of the intrinsic bias (blue curve) is kept at the home base location on the floor while the vertical

*Figure 5 continued on next page*

*Figure 5 continued*

coordinate travels along with the observer. Thus, upon stepping down from the stepladder and standing on the floor, the observer underestimates the horizontal distance of the target. The target underestimation would be similar to that of the *horizontal-walking* condition depicted in **b**. In contrast, if the path-integration process integrates distance in both the horizontal and vertical directions, the intrinsic bias will remain at the top of the staircase (orange curve). The perceived target location will then be dramatically different from that in the *horizontal-walking* condition. (**b**) *Horizontal-walking* condition. (**c**) The average results of the *baseline-stationary*, *horizontal-walking*, and *stepladder* conditions, respectively, represented by the red circle, green square, and blue triangle symbols (*n* = 9). The judged horizontal distances are significantly shorter in the *horizontal-walking* condition than in the *baseline-stationary* conditions. The two sets of data points are fitted by the same intrinsic bias profile with a horizontal separation of 1.0 m, which is close to the walked distance (1.06 m) from the home base to the new location in the *horizontal-walking* condition. Of significance, the judged locations in the *stepladder* condition (blue triangles) are similar to that in the *horizontal-walking* condition (green squares). This confirms that the path-integration process mainly gauges the horizontal (ground) distance traveled. (**d**) Average judged angular declination as a function of the physical angular declination. Error bars represent standard errors of the mean (n=9).

The online version of this article includes the following source data for figure 5:

**Source data 1.** Judegd locations of Experiment 5.

- Interaction effect: $F_{(5, 35)} = 1.072$, p = 0.392

## Experiment 3 (*Figure 4*)

### Vestibular-forward condition vs. walking condition (Figure 4a)

We applied two-way ANOVA with repeated measures (2 test conditions × 6 horizontal distances) to the walked horizontal distance data. The analysis reveals:

- Main effect of test condition (*vestibular-forward vs. walking*): $F_{(1, 7)} = 0.142$, p = 0.718
- Main effect of horizontal distance: $F_{(5, 35)} = 99.788$, p < 0.001
- Interaction effect: $F_{(5, 35)} = 0.714$, p = 0.617

### Vestibular-forward condition vs. baseline-stationary condition (Figure 4a)

We applied two-way ANOVA with repeated measures (2 test conditions × 6 horizontal distances) to the walked horizontal distance data. The analysis reveals:

- Main effect of test condition (*baseline-stationary vs. vestibular-forward*): $F_{(1, 7)} = 598.551$, p < 0.001
- Main effect of horizontal distance: $F_{(5, 35)} = 168.855$, p < 0.001
- Interaction effect: $F_{(5, 35)} = 19.422$, p < 0.001

### Baseline-stationary condition vs. walking condition (Figure 4a)

We applied two-way ANOVA with repeated measures (2 test conditions × 6 horizontal distances) to the walked horizontal distance data. The analysis reveals:

- Main effect of test condition (*baseline-stationary vs. walking*): $F_{(1, 7)} = 299.648$, p < 0.001
- Main effect of horizontal distance: $F_{(5, 35)} = 160.078$, p < 0.001
- Interaction effect: $F_{(5, 35)} = 12.607$, p < 0.001

### Vestibular-backward condition vs. baseline-stationary condition (Figure 4b)

We applied two-way ANOVA with repeated measures (2 test conditions × 6 horizontal distances) to the walked horizontal distance data. The analysis reveals:

- Main effect of test condition (*vestibular-backward vs. baseline-stationary*): $F_{(1, 7)} = 651.863$, p < 0.001
- Main effect of horizontal distance: $F_{(5, 35)} = 51.558$, p < 0.001
- Interaction effect: $F_{(5, 35)} = 10.848$, p < 0.001

## Experiment 4 (*Figure 4e*)

### Vestibular-forward condition vs. baseline-stationary condition (verbal reports of eye-to-target distances)

We applied two-way ANOVA with repeated measures (2 test conditions × 5 horizontal distances) to the verbally reported distance data. The analysis reveals:

- Main effect of test condition (*vestibular-forward vs. baseline-stationary*): $F(1, 7) = 217.749$, $p < 0.001$
- Main effect of distance: $F(4, 28) = 36.945$, $p < 0.001$
- Interaction effect: $F(4, 28) = 8.973$, $p < 0.001$

## Experiment 5 (*Figure 5c*)

### Baseline-stationary condition vs. stepladder condition

#### Horizontal distances

We applied two-way ANOVA with repeated measures (2 test conditions × 5 horizontal distances) to the walked horizontal distance data. The analysis reveals:

- Main effect of test condition (*baseline-stationary vs. stepladder*): $F(1, 8) = 153.541$, $p = 0.000$
- Main effect of horizontal distance: $F(4, 32) = 35.954$, $p = 0.000$
- Interaction effect: $F(2.58, 20.69) = 3.89$, $p = 0.028$

#### Heights

We applied two-way ANOVA with repeated measures (2 test conditions × 5 heights) to the judged height data. The analysis reveals:

- Main effect of test condition (*baseline-stationary vs. stepladder*): $F(1, 8) = 0.112$, $p = 0.747$
- Main effect of height: $F(4, 32) = 181.666$, $p = 0.000$
- Interaction effect: $F(4, 32) = 0.288$, $p = 0.883$

### Baseline-stationary condition vs. horizontal-walking condition

#### Horizontal distances

We applied two-way ANOVA with repeated measures (2 test conditions × 5 horizontal distances) to the walked horizontal distance data. The analysis reveals:

- Main effect of test condition (*baseline-stationary vs. horizontal-walking*): $F(1, 8) = 104.198$, $p = 0.000$
- Main effect of horizontal distance: $F(4, 32) = 45.768$, $p = 0.000$
- Interaction effect: $F(1.892, 15.137) = 8.461$, $p = 0.004$

#### Heights

We applied two-way ANOVA with repeated measures (2 test conditions × 5 heights) to the judged height data. The analysis reveals:

- Main effect of test condition (*baseline-stationary vs. horizontal-walking*): $F(1, 8) = 7.695$, $p = 0.024$
- Main effect of height distance: $F(4, 32) = 236.530$, $p = 0.000$
- Interaction effect: $F(2.636, 21.088) = 2.472$, $p = 0.096$

### Stepladder condition vs. horizontal-walking condition

#### Horizontal distances

We applied two-way ANOVA with repeated measures (2 test conditions × 5 horizontal distances) to the walked horizontal distance data. The analysis reveals:

- Main effect of test condition (*stepladder vs. horizontal-walking*): $F(1, 8) = 0.532$, $p = 0.487$
- Main effect of horizontal distance: $F(4, 32) = 32.372$, $p = 0.000$
- Interaction effect: $F(2.967, 23.737) = 0.874$, $p = 0.467$

### Heights

We applied two-way ANOVA with repeated measures (2 test conditions × 5 heights) to the judged height data. The analysis reveals:

- Main effect of test condition (*stepladder vs. horizontal-walking*): $F(1, 8) = 0.864$, p = 0.380
- Main effect of height: $F(4, 32) = 185.248$, p = 0.000
- Interaction effect: $F(2.864, 22.908) = 0.829$, p = 0.487

## Discussion

We constantly move relative to the static environment, which causes our retinal images of the environment to also be in constant motion. Yet, we reliably localize objects as we navigate our environment. This raises the question of how our visual system creates a stable visual space for us to operate and plan future actions, such as walking, despite the inadequate retinal inputs. While it was known the visual system relies predominantly on the ground surface to accurately localize objects, it was unclear what coordinate system it uses for coding visual space. In this study, we proposed the visual system uses an allocentric, or world-centered, coordinate system for the ground-based reference. In this way, the visual system could alleviate the challenge of constant retinal motion during self-motion by streamlining online location computation to only the moving observer and the object of interest. The locations of surrounding objects are anchored to a fixed position on the ground negating excessive computation, thereby reducing coding redundancy. Our main finding provides support for the allocentric hypothesis. We showed during self-motion, the intrinsic bias that acts as the spatial reference of the ground surface when visual cues are not visible, is fixed to the ground location before the motion begins (home base) and remains at the same ground location during self-motion.

Previous studies mainly tested location judgments of observers who stood stationary at one location on the ground while viewing a stationary target (similar to our *baseline-stationary* condition). Because the observers need not generate significant self-motion in those studies, it was not possible to investigate the allocentric hypothesis. The previous findings with stationary observers showed that the visual system uses the ground surface representation as a reference frame for representing object locations (*Gibson, 1950*; *Sedgwick, 1986*; *Sinai et al., 1998*; *Meng and Sedgwick, 2001*; *Ooi et al., 2001*; *Wu et al., 2004*; *He et al., 2004*; *Bian et al., 2005*; *Philbeck and Loomis, 1997*; *Gogel and Tietz, 1979*). One may thu s claim that the ground-based spatial coordinate system is world-centered or allocentric. However, it is hard to reject the egocentric hypothesis since the origin of the spatial coordinate system is attached to the surface underneath one's feet while standing still on the ground. In a way, the egocentric hypothesis is implicitly assumed because it is less complex than the allocentric hypothesis. After all, without body motion, the allocentric coordinate system does not have an advantage over the egocentric coordinate system in terms of processing efficiency.

The visual system relies on the path-integration mechanism to ground the intrinsic bias at the home base. The path-integration mechanism can function when the observer's motion is passive, suggesting the traveled distance can be based on the vestibular signals alone. Also, the path-integration mechanism fails to operate efficiently when the observer's attention is distracted to an unrelated cognitive demand during walking. This suggests some degree of attentional effort is required to reliably path integrate an observer's body location relative to the environment. Lastly, we showed the path-integration mechanism is more efficient in the horizontal than the vertical direction (assymetric with a ground bias). This observation further highlights one of the most important design principles underlying our sensory and motor systems, namely, fitting our terrestrial niche. We inhabit a land-based environment and most of our daily activities are structured relative to the ground. The current finding on the vertical–horizontal asymmetry of the path-integration mechanism for visual space perception is consistent with previous studies on spatial memory, which revealed the path-integration systems of land-dwelling animals are more efficient in estimating horizontally traveled distance (*Wohlgemuth et al., 2001*; *Jovalekic et al., 2011*; *Hayman et al., 2011*; *Zwergal et al., 2016*; *Ronacher, 2020*). In particular, our observation is reminiscent of the behavioral studies of the desert ants, *Cataglyphis fortis*. *Wohlgemuth et al., 2001* found desert ants used path integration to return from foraging excursions on a shortcut way to their nests (also see *Ronacher, 2020*). Specifically, it was revealed when the ants were crawling over a slope surface with uphill and downhill trajectory, their path-integration system computed only the horizontal component of the traveled distance. We note, however, further research

is needed to investigate whether the same path-integration mechanism is employed for a spatial memory task (previous studies by others) and a spatial perception task (current study).

Overall, our study suggests the visual system can create a world-centered visual space by anchoring its spatial reference frame to the ground during navigation. One advantage of adopting this space coding strategy is that when we travel, the spatial representations of most surrounding objects, which are static relative to the ground, will remain constant (static) despite the retinal image motion. Accordingly, our visual system only needs to dynamically update the spatial representations of our bodies and the few objects that are moving relative to an allocentric origin on the ground (within a world-centered context). A consequential byproduct of operating within the framework of an allocentric, world-centered visual space is the perceived stability of our visual environment during locomotion.

Let us consider the following hypothetical situation of a forward walking condition in the dark. The observer sees a target briefly before walking forward over a short distance, and again after they reach the new destination. *Figure 2d and e* illustrate the perceived target locations, respectively, if the visual system adopts the allocentric or egocentric coding strategy. Noticeably, the perceived target locations differ before (open red circle) and after walking (closed circle) as shown in both figures. However, the separation between the perceived target locations is smaller in *Figure 2d* (allocentric coding) than *Figure 2e* (egocentric coding), indicating more perceived location stability in the former. Although, this position stability has an accuracy trade-off as the perceived target locations after the short walk are more underestimated in *Figure 2d* (allocentric) than *Figure 2e* (egocentric). Nevertheless, there is reason to believe that the location inaccuracy from using the allocentric spatial coding strategy will be minimized in the full cue environment, where the influence of the intrinsic bias is less. This is because when the ground is not visible in the dark, the surface slant of the intrinsic bias is responsible for the distance underestimation of a target on the ground (*Figure 1c*). When there is some sparse texture information on the ground (*Figure 1d*), the ground surface representation becomes less slanted than the intrinsic bias and the distance underestimation error is less. Accordingly, the underestimation error due to using the allocentric spatial coding strategy will be further reduced (unpublished data). Therefore, a full cue condition with rich texture gradient information on the ground will render the ground surface representation largely veridical, resulting in accurate distance perception.

The observations reported here support the proposal that during observer self-motion the brain establishes an allocentric intrinsic bias by employing the path-integration mechanism. By utilizing the allocentric intrinsic bias, the visual system is able to construct an allocentric ground reference frame. Accordingly, the ground-based visual space can effectively program and guide navigation of terrestrial creatures on the ground surface. Given its significance, it is natural to ask where in the brain the intrinsic bias, a prototype model of the ground surface, is implemented. In this respect, we are reminded that the two-dimensional cognitive map (long-term spatial memory) employed by land-dwelling animals is also linked to the terrain where they travel (*Jovalekic et al., 2011*; *Hayman et al., 2011*; *Tolman, 1948*; *O'Keefe and Dostrovsky, 1971*; *O'Keefe and Nadel, 1978*; *Moser et al., 2014*). Therefore, it is reasonable to speculate that the intrinsic bias could be a function of the mammalian hippocampal formation that is well known to be responsible for directing navigation when spatial sensory inputs are impoverished or lacking such as in the dark. Consistent with this speculation, *Nau et al., 2018* pointed to growing evidence that 'the mammalian hippocampal formation, extensively studied in the context of navigation and memory, mediates a representation of visual space that is stably anchored to the external world' (*Nau et al., 2018*).

Following this line of thinking, we further speculate that the intrinsic bias that represents the internal model of the visual ground surface is implemented in the spatial memory system. As we mentioned earlier, our visual system employs the ground surface representation as a spatial reference frame for spatial coding, fitting our terrestrial niche. It is thus tempting to adopt the idea that the spatial memory systems of land dwellers employ the same spatial coding, with the cognitive map serving as an internal model of the ground and the associated spatial features (walls and obstacles) that are supported by the ground. As such, the intrinsic bias is the basis not only for the cognitive map within the medial temporal lobe but also for visual ground surface representation within the perceptual cortex (*Byrne et al., 2007*; *Bottini and Doeller, 2020*; *Wang et al., 2020*). Future investigations should explore the broader notion that the concept of the intrinsic bias could be genetically or epigenetically coded, serving as an important priori knowledge for the brain to represent the ground surface and to construct our inner spatial world (*Gibson, 1950*; *Kant, 1963*; *Von Uexküll, 2013*).

# Materials and methods

## Observers

Fourteen observers (age = 24.47 ± 1.45 years old; eye height = 1.60 ± 0.02 m; 7 males and 7 females) participated in Experiments 1–4 with eight observers for each experiment. Nine observers (age = 22.78 ± 1.22 years old; eye height = 1.58 ± 0.04 cm; 6 males and 3 males) participated in Experiment 5. They were naïve to the purpose of the study. All observers had normal, or corrected-to-normal, visual acuity (at least 20/20) and a stereoscopic resolution of 20 arc sec or better. They viewed the visual scene binocularly. A within-subject experimental design was used. The study protocol was approved by The University of Louisville Institutional Review Board (approved IRB number 94.0302) and followed the tenets of the Declaration of Helsinki. All subjects signed the informed consent form approved by the IRB at the start of the study.

## General stimulus and testing environment

All Experiments 1–4 were performed in a dark room whose layout and dimensions were unknown to the observers. One end of the room, just before the testing area, served as the waiting area (~3 m²) for the observer. The waiting area had a chair facing the wall for the observer to sit in between trials so that his/her back faced the testing area. Two white LED lights on the ceiling provided ambient illumination while the observer waited in between trials. The testing and waiting areas were separated by a black curtain. A long guidance rope (0.8 m above the floor) was tied to both ends of the room and served to guide the observer while walking blindly. A plastic wrap was tied to the guidance rope on the part of the rope located in the waiting area near the curtain to mark the start point (home base). To ready for a trial, the observer walked to this start point and faced the test area while holding onto the plastic wrap on the rope and called out 'ready'. The curtain was then drawn open for the trial to begin.

The dimly lit test target used in all experiments was a diffused green LED (0.16 cd m⁻²). The LED was placed in the center of a ping-pong ball that was encased in a small opaque box. An adjustable iris-diaphragm aperture was placed at the front of the ping-pong ball to keep its visual angular size constant at 0.22° when measured at the eye level. During testing, the target was displayed with a 5-Hz flicker for 1 s. Music was played aloud during the entire experimental session to mask extraneous auditory information during the experiments.

## Observer's response tasks

The main task used in all experiments was the *blind walking-gesturing task* (*Ooi et al., 2001*; *Ooi et al., 2006*). For each trial, the observer stood by the guidance rope in the dark and judged the location of the 1-s flickering target. After which he/she then put on the blindfold and called out 'ready to walk'. This signaled the experimenter to quickly remove the target and shook the guidance rope to indicate it was safe to walk. The observer walked while sliding his/her right hand along the guidance rope until he/she reached the remembered target location. Once there, he/she indicated the remembered target height with his/her left hand and called out 'done'. The experimenter turned on the flashlight, marked the observer's feet location, measured the gestured height, and asked the observer to turn around and walk back to the start point using the guidance rope. When the observer arrived at the start point, the experimenter turned on the ceiling LED lights in the waiting area. The observer then removed the blindfold, sat down, and waited for the next trial.

An additional task used, in Experiment 4, was the *verbal reporting task* (*Ooi et al., 2001*; *Zhou et al., 2013*). Here, the observer stood next to the guidance rope in the dark and viewed the target for 5 s to judge its absolute distance between the target and his/her eyes. Once the target was turned off, an audio tone was presented to signal the observer to immediately report the estimated distance either in meters or feet. For both this and the blind walking-gesturing task, the observer was provided five practice trials before each test session. No feedback regarding performance was provided to the observer during the practice or test session.

## Experiment 1

### Design

Two viewing conditions were tested: *walking* and *baseline-stationary*. Each condition was tested separately with a 12- and 60-s waiting period. The target was placed at one of six locations in both

conditions. Four locations were on the floor at 3, 4.25, 5.25, or 6.5 m from the observer, the fifth location was 0.5 m above the floor at 6.5 m from the observer, and the sixth location was at the observer's eye level and 6.5 m from the observer. Testing of the 4.25, 5.25, and 6.5 m targets on the floor were repeated three times while testing of the remaining targets was repeated twice. A total of 60 trials were run over 2 days. The order of stimulus presentation was randomized.

### Procedure

While the observer sat at the waiting area before each trial, he/she was informed of the upcoming test condition (*baseline-stationary* or *walking*) and of the waiting period (12 or 60 s). After that, an audio tone was presented to signal to the observer to walk to the start point (home base) and face the black curtain in the direction of the testing area. About 30 s later, the experimenter turned off the ambient LED lights in the waiting area and the observer drew the curtain open in the dark.

For the *baseline-stationary* condition, the observer stood in the dark at the start point (home base) over the predetermined waiting duration (12 or 60 s). He/she was instructed to stand upright with minimal head motion during the waiting period, and to expect hearing a pure tone at the end of the waiting period. Roughly, 2 s after hearing the tone, the test target was presented at one of the six predetermined locations. The observer's task was to judge its location and perform the blind walking-gesturing task. For the *walking* trial, the observer stood at the start point until a white noise (instead of pure tone) was heard. He/she then walked forward until his/her right hand touched a plastic wrap tied on the guidance rope at the new location (1.5 m from the start point). He/she then stopped walking, called out 'ready' and waited there for either 12 or 60 s before the test target was presented. The remaining procedural steps were the same as in the *baseline-stationary* condition.

## Experiment 2
### Design and procedure

Both the *baseline-stationary* and *walking* conditions with the 12-s waiting period were tested but with one modification. The modification was that during the 12-s waiting period at the start point (home base) (for *baseline-stationary* condition), or while walking to, and waiting at the new location (for *walking* condition), the observer performed a counting task. The experimenter would provide a random number between 50 and 99 for the observer to count backward. The same six target locations tested in Experiment 1 were tested here. A total of 30 trials were run. The order of stimulus presentation was randomized.

## Experiment 3
### Vestibular-forward condition
#### Design

The two conditions from Experiment 1 (*walking* and *baseline-stationary* conditions with the 12-s waiting period) and a new, *vestibular-forward* condition were tested. The latter *vestibular-forward* condition was the same as in the *walking* condition, except that the observer now stood on a rolling-platform that was pushed forward by the experimenter for 1.5 m. The rolling-platform had four wheels that was 0.30 m above the floor. The same six target locations tested in Experiment 1 were tested here. There were 60 trials in total with 30 trials tested per day. The order of stimulus presentation was randomized.

#### Procedure

Two experimenters (A and B) conducted the experiment. Before each trial, Experimenter A instructed the observer of the upcoming test condition and to prepare for an audio signal. Once the signal was heard, the observer stepped onto the rolling-platform at the start point (home base) behind the curtain with eyes opened and facing the test area. He/she then called out 'ready'. A second audio signal was presented after 30 s. A pure tone indicated to the observer to stay (*baseline-stationary* condition), while a white noise indicated to walk (*walking* condition) or to be pushed forward (*vestibular-forward* condition). At the same time, Experimenter A turned off the LED lights in the waiting area, and the observer drew the curtain open. For the *baseline-stationary* condition trial, the observer stepped down from the rolling-platform, stood at the start point, and called out 'ready to view'. For the *walking*

condition trial, the observer stepped down from the rolling-platform, and walked forward until he/she touched the plastic wrap tied onto the guidance rope at the new location (1.5 m), and then stopped and called out 'ready to view'. For the *vestibular-forward* condition trial, the observer kept standing on the rolling-platform. Experimenter B verbally informed the observer the trial was starting and pushed the rolling-platform forward by 1.5 m to the new location. Upon arrival, Experiment B instructed the observer to step down from the rolling-platform and to call out 'ready to view'. For all three trial types, after the observer called out 'ready to view', Experimenter A turned on the test target after a 12-s waiting period for the observer to judge its location. He/she then responded by performing the blind walking-gesturing task.

### Vestibular-backward condition
#### Design and procedure
The *baseline-stationary* condition was the same as in Vestibular-forward condition. The *vestibular-backward* condition was modified from the *vestibular-forward* in Vestibular-forward condition, by pulling the rolling-platform backward for 1.5 m. The same six target locations as in Vestibular-forward condition were tested. A total of 30 trials were tested in one session. The order of stimulus presentation was randomized.

## Experiment 4
### Design
The two conditions tested were the *vestibular-forward* and *baseline-stationary* conditions as in Vestibular-forward condition. But instead of the six test target locations, only five target locations on the floor were tested. The target location at 6.5 m distance and 0.5 m above the ground was not tested. All targets were presented for 5 s. Twenty-eight trials in total were tested in one session. The order of stimulus presentation was randomized.

### Procedure
The procedure was the same as in Vestibular-forward condition, except the observers performed the verbal report task. To do so, the observer judged the absolute distance between the target and his/her eyes. Immediately after the target was turned off, an audio signal was presented to prompt the observer to report the estimated distance aloud either in meters or feet.

## Experiment 5
### Design
Three conditions (*baseline-stationary*, *horizontal-walking*, and *stepladder*) were tested in two different blocks. The *horizontal-walking* and *baseline-stationary* conditions were mixed in test block-A, while the *stepladder* and *baseline-stationary* conditions in test block-B. Each block consisted of 20 test trials. The order of test conditions within each block was randomized. Each block was tested in a daily session and ran twice. The testing order of the four blocks was alternated between observers.

### Procedure
Before each trial, the experimenter informed the observer of the condition to be tested. The *baseline-stationary* and *horizontal-walking* conditions were conducted as in the main experiment (Experiment 1). For the *stepladder* condition, the observer first ascended the stepladder, and waited for an audio tone that signaled to descend the stepladder with eyes opened and looking at the invisible horizon (in the dark). Upon reaching the foot of the stepladder, he/she held onto the guidance rope and walked forward until he/she felt the plastic wrap on the rope at the new location. The observer then stood still and waited for 12 s before the test target was presented. After judging the target's location, he/she put on the blindfold and called out 'ready' to begin the blindfolded walking-gesturing task.

### Statistical tests
Data were analyzed using ANOVA with repeated measures. The Mauchly's test was applied to verify the assumption of sphericity.

## Acknowledgements

We thank Dr. Jack Loomis for the helpful discussions and feedback on an earlier draft of the current paper. The work was supported by a grant from the National Institutes of Health (EY033190) to ZJH and TLO. The funders had no role in study design, data collection, and analysis, decision to publish, or preparation of the manuscript.

## Additional information

### Funding

| Funder | Grant reference number | Author |
|--------|------------------------|--------|
| National Eye Institute | EY 033190 | Teng Leng Ooi<br>Zijiang J He |

The funders had no role in study design, data collection, and interpretation, or the decision to submit the work for publication.

### Author contributions

Liu Zhou, Data curation, Software, Formal analysis, Validation, Investigation, Visualization, Methodology, Writing – original draft, Writing – review and editing; Wei Wei, Software, Investigation, Methodology; Teng Leng Ooi, Zijiang J He, Conceptualization, Resources, Formal analysis, Supervision, Funding acquisition, Validation, Visualization, Methodology, Writing – original draft, Project administration, Writing – review and editing

### Author ORCIDs

Zijiang J He ⬦ https://orcid.org/0000-0001-6313-9016

### Ethics

The study protocol was approved by the The University of Louisville Institutional Review Board (approved IRB number 94.0302) and followed the tenets of the Declaration of Helsinki. All subjects signed the informed consent form approved by the IRB at the start of the study.

Reviewer #1 (Public Review): https://doi.org/10.7554/eLife.88095.3.sa1
Reviewer #3 (Public Review): https://doi.org/10.7554/eLife.88095.3.sa2
Author response https://doi.org/10.7554/eLife.88095.3.sa3

## Additional files

### Supplementary files

• MDAR checklist

### Data availability

All data are available in the main text. The data sets generated are available from OSF.

The following dataset was generated:

| Author(s) | Year | Dataset title | Dataset URL | Database and Identifier |
|-----------|------|---------------|-------------|-------------------------|
| He Z | 2024 | An allocentric human odometer for perceiving distances on the ground plane | https://osf.io/afrk9/ | Open Science Framework, afrk9 |

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
