## [Editor Report · eLife assessment]

This **important** study reveals the use of an allocentric spatial reference frame in the updating perception of the location of a dimly lit target during locomotion. The evidence supporting this claim is **compelling**, based on a series of cleverly and carefully designed behavioral experiments. The results will be of interest not only to scientists who study perception, action and cognition but also to engineers who work on developing visually guided robots and self-driving vehicles.

---

## [Referee Report · Reviewer #1 (Public Review)]

This study conducted a series of experiments to comprehensively support the allocentric rather than egocentric visual spatial reference updating for the path-integration mechanism in the control of target-oriented locomotion. Authors firstly manipulated the waiting time before walking to tease apart the influence from spatial working memory in guiding locomotion. They demonstrated that the intrinsic bias in perceiving distance remained constant during walking and that the establishment of a new spatial layout in the brain took a relatively longer time beyond the visual-spatial working memory. In the following experiments, the authors then uncovered that the strength of the intrinsic bias in distance perception along the horizontal direction is reduced when participants' attention is distracted, implying that world-centered path integration requires attentional effort. This study also revealed horizontal-vertical asymmetry in a spatial coding scheme that bears a resemblance to the locomotion control in other animal species such as desert ants.

The revised version of the study effectively situates the research within the broader context of terrestrial navigation, focusing on the movement of land-based creatures and offers a clearer explanation for the potential neurological basis of the human brain's allocentric odometer. Previous feedback has been thoroughly considered, and additional details have been incorporated into the presentation of the results.

---

## [Referee Report · Reviewer #3 (Public Review)]

This study investigated what kind of reference (allocentric or egocentric) frame we used for perception in darkness. This question is essential and was not addressed much before. The authors compared the perception in the walking condition with that in the stationary condition, which successfully separated the contribution of self-movement to the spatial representation. In addition, the authors also carefully manipulated the contribution of the waiting period, attentional load, vestibular input, testing task, and walking direction (forward or backward) to examine the nature of the reference frame in darkness systematically.

I am a bit confused by Figure 2b. Allocentric coordinate refers to the representation of the distance and direction of an object relative to other objects but not relative to the observer. In Figure 2, however, the authors assumed that the perceived target was located on the interception between the intrinsic bias curve and the viewing line from the NEW eye position to the target. This suggests that the perceived object depends on the observer's new location, which seems odd with the allocentric coordinate hypothesis.

According to Fig 2b, the perceived size should be left-shifted and lifted up in the walking condition compared to that in the stationary condition. However, in Figure 3C and Fig 4, the perceived size was the same height as that in the baseline condition.

Is the left-shifted perceived distance possibly reflecting a kind of compensation mechanism? Participants could not see the target's location but knew they had moved forward. Therefore, their brain automatically compensates for this self-movement when judging the location of a target. This would perfectly predict the left-shifted but not upward-shifted data in Fig 3C. A similar compensation mechanism exists for size constancy in which we tend to compensate for distance in computing object size.

According to Fig 2a, the target, perceived target, and eye should be aligned in one straight line. This means that connecting the physical targets and the corresponding perceived target results in straight lines that converge at the eye position. This seems, however, unlikely in Figure 3c.

---

## [Author Response]

The following is the authors’ response to the original reviews.

**Public Reviews:**

**Reviewer 1:**
(1) Authors need to acknowledge the physical effort in addition to visual information for the spatial coding and may consider the manipulation of physical efforts in the future to support the robustness of constant intrinsic bias in ground-based spatial coding during walking.

Whether one’s physical effort can affect spatial coding for visual perception is not a settled issue. Several empirical studies have not been able to obtain evidence to support the claim. For example, empirical studies by Hutchison & Loomis (2009) and Durgin et al. (2009) did not find wearing a heavy backpack significantly influenced distance perception, in contrast to the findings by Proffitt et al (2003). We respectfully request not to discuss this issue in our revision since it is not closely related to the focus of the current study.

(2) Furthermore, it would be more comprehensive and fit into the Neuroscience Section if the authors can add in current understandings of the spatial reference frames in neuroscience in the introduction and discussion, and provide explanations on how the findings of this study supplement the physiological evidence that supports our spatial perception as well. For instance, world-centered representations of the environment, or cognitive maps, are associated with hippocampal formation while self-centered spatial relationships, or image spaces, are associated with the parietal cortex (see Bottini, R., & Doeller, C. F. (2020). Knowledge Across Reference Frames: Cognitive Maps and Image Spaces. Trends in Cognitive Sciences, 24(8),606-619. https://doi.org/10.1016/j.tics.2020.05.008 for details)

We have now added this important discussion in the revision on pages 12-13.

We thank the reviewer for the helpful comments.

**Reviewer 2:**
(1) ….As a result, it is unclear to what extent this "allocentric" intrinsic bias is involved in our everyday spatial perception. To provide more context for the general audience, it would be beneficial for the authors to address this issue in their discussion.

We have clarified this on pages 3-4. In brief, our hypothesis is that during self-motion, the visual system constructs an allocentric ground surface representation (reference frame) by integrating the allocentric intrinsic bias with the external depth cues on the natural ground surface. Supporting this hypothesis, we recently found that when there is texture cue on the ground, the representation of the ground surface is influenced by the allocentric intrinsic bias (Zhou et al, unpublished results).

(2) The current findings on the "allocentric" coding scheme raise some intriguing questions as to why such a mechanism would be developed and how it could be beneficial. The finding that the "allocentric" coding scheme results in less accurate object localization and requires attentional resources seems counterintuitive and raises questions about its usefulness. However, this observation presents an opportunity for the manuscript to discuss the potential evolutionary advantages or trade-offs associated with this coding mechanism.

The revision has discussed these important issues on page 12.

(3) The manuscript lacks a thorough description of the data analysis process, particularly regarding the fitting of the intrinsic bias curve (e.g., the blue and gray dashed curve in Figure 3c) and the calculation of the horizontal separation between the curves. It would be beneficial for the authors to provide more detailed information on the specific function and parameters used in the fitting process and the formula used for the separation calculation to ensure the transparency and reproducibility of the study's results.

The results of the statistical analysis were presented in the supplementary materials. We had stated in the original manuscript that we fitted the intrinsic bias curve by eye (obtained by drawing the curve to transcribe the data points as closely as possible) (page 26). This is because we do not yet have a formula for the intrinsic bias. A challenge is the measured intrinsic bias in the dark can be affected by multiple factors. One factor is related to individual differences as the intrinsic bias is shaped by the observer’s past experiences and their eye height relative to the ground surface. However, it is certainly our goal to develop a quantitative model of the intrinsic bias in the future.

We thank the reviewer for the helpful comments.

**Reviewer 3:**
(1) I am a bit confused by Figure 2b. Allocentric coordinate refers to the representation of the distance and direction of an object relative to other objects but not relative to the observer. In Figure 2, however, the authors assumed that the perceived target was located on the interception between the intrinsic bias curve and the viewing line from the NEW eye position to the target. This suggests that the perceived object depends on the observer's new location, which seems odd with the allocentric coordinate hypothesis.

We respectively disagree with the Reviewer’s statement that “Allocentric coordinate refers to the representation of the distance and direction of an object relative to other objects but not relative to the observer.” The statement conflates the definitions of allocentric representation with exocentric representation. We respectfully maintain that the observer’s body location, as well as observer-object distance, can be represented with the allocentric coordinate system.

(2) According to Fig 2b, the perceived size should be left-shifted and lifted up in the walking condition compared to that in the stationary condition. However, in Figure 3C and Fig 4, the perceived size was the same height as that in the baseline condition.

We assume by “target size”, the Reviewer actually meant, “target location”. It is correct that figure 3c and figure 4 showed judged distance changed as predicted, while the change in judged height was not significant. One explanation for this is that the magnitude of the height change was much smaller than the distance change and could not be revealed by our blind walking-gesturing method. Please also note our figures used difference scales for the vertical height and horizontal distance.

(3) Is the left-shifted perceived distance possibly reflecting a kind of compensation mechanism? Participants could not see the target's location but knew they had moved forward. Therefore, their brain automatically compensates for this self-movement when judging the location of a target. This would perfectly predict the left-shifted but not upward-shifted data in Fig 3C. A similar compensation mechanism exists for size constancy in which we tend to compensate for distance in computing object size.

We assume the Reviewer suggested that the path-integration mechanism first estimates the traveled distance in the dark, and then the brain subtracts the estimated distance from the perceived target distance. We respectfully maintain that this explanation is unlikely because it does not account for our empirical findings. We found that walking in the dark did not uniformly affect perceived target distance, as the Reviewer’s explanation would predict. As shown in figures 3 and 4, walking affected the near targets less than the far targets (i.e., the horizontal distance difference between walking and baseline-stationary conditions was smaller for the near target than far target).

(4) According to Fig 2a, the target, perceived target, and eye should be aligned in one straight line. This means that connecting the physical targets and the corresponding perceived target results in straight lines that converge at the eye position. This seems, however, unlikely in Figure 3c.

We have added in the revision, the averaged eye positions on the y-axes of figures 3 and 4. To reveal the impact of the judged angular declination, we also added graphs that plotted the estimated angular declination as a function of the physical declination of the target. In general, the slopes are close to unity.

We thank the reviewer for the helpful comments.

**Recommendations for the authors:**

**Reviewer 1 (Recommendations For The Authors):**
(1) This study is very well-designed and written. One minor comment is that anisotropy usually refers to the perceptual differences along cardinal (horizontal + vertical) and oblique directions. It might be clearer if the authors changed the "horizontal-vertical anisotropy" to "horizontal/vertical asymmetry”.

The Reviewer is correct, and we have changed it to horizontal/vertical asymmetry (pages 8 and 11).

**Reviewer 2 (Recommendations For The Authors):**
(1) Providing more details about the "path integration mechanism" when it is first introduced in line 44 would be helpful for readers to better understand the concept.

The revision has expanded on the path integration mechanism (page 4).

Adding references for the statement starting with "In fact, previous findings" in lines 218 and would be helpful to provide readers with a basis for comparison between the current study and previous studies that reported an egocentric coding system.

We have added the references and elaborated on this important issue (pages 10-11).

(2) There appears to be a discrepancy between the Materials and Methods section, which states that 14 observers participated in Experiments 1-4, and the legends of Figures 3 and 4, which indicates a sample size of "n=8." It would be helpful if the authors could clarify this discrepancy and provide an explanation for the difference in the sample size reported.

We have clarified the number of observers on page 14.

(3) While reporting statistical significance is essential in the Results section, there are several instances where the manuscript only mentions a "statistically significant separation" with it p-value without providing the mean and standard deviation of the separation values (e.g., line 100 and 120). This can make it difficult for readers to fully grasp the quantitative nature of the results.

The statistical analysis and outcomes were presented in the supplementary information document in our original submission.

**Reviewer 3 (Recommendations For The Authors):**
(1) Figure 1 is not significantly related to the current manuscript.

We feel that retaining figure 1 in the manuscript would help readers to quickly grasp the background literature without having to refer extensively to our previous publications.

(2) Add eye position to the results figures.

We have added eye positions in the figures.

(3) Fig 4c requires a more detailed explanation. The authors stated that Figures 4a and 4c showed consistent results. However, because 4a and 4c used different horizontal axis, it is different to compare them directly.

We have modified the sentence in the revision (page 8).